# Reversibly Migratable Fluorescent Probe for Precise and Dynamic Evaluation of Cell Mitochondrial Membrane Potentials

**DOI:** 10.3390/bios12100798

**Published:** 2022-09-27

**Authors:** Guofen Song, Haiwei He, Wanling Chen, Yuanliang Lv, Paul K. Chu, Huaiyu Wang, Penghui Li

**Affiliations:** 1Institute of Biomedicine and Biotechnology, Shenzhen Institute of Advanced Technology, Chinese Academy of Sciences, Shenzhen 518055, China; 2University of Chinese Academy of Sciences, Beijing 100049, China; 3Department of Physics, City University of Hong Kong, Tat Chee Avenue, Kowloon, Hong Kong 999077, China; 4Department of Materials Science and Engineering, City University of Hong Kong, Tat Chee Avenue, Kowloon, Hong Kong 999077, China; 5Department of Biomedical Engineering, City University of Hong Kong, Tat Chee Avenue, Kowloon, Hong Kong 999077, China

**Keywords:** reversible, migratable, mitochondrial membrane potential, fluorescent probe, dynamic evaluation

## Abstract

The mitochondrial membrane potential (MMP, ΔΨ_mito_) provides the charge gradient required for mitochondrial functions and is a key indicator of cellular health. The changes in MMP are closely related to diseases and the monitoring of MMP is thus vital for pathological study and drug development. However, most of the current fluorescent probes for MMP rely solely on the cell fluorescence intensity and are thus restricted by poor photostability, rendering them not suitable for long-term dynamic monitoring of MMP. Herein, an MMP-responsive fluorescent probe pyrrolyl quinolinium (**PQ**) which is capable of reversible migration between mitochondria and nucleolus is developed and demonstrated for dynamic evaluation of MMP. The fluorescence of **PQ** translocates from mitochondria to nucleoli when MMP decreases due to the intrinsic RNA-specificity and more importantly, the translocation is reversible. The cytoplasm to nucleolus fluorescence intensity ratio is positively correlated with MMP so that this method avoids the negative influence of photostability and imaging parameters. Various situations of MMP can be monitored in real time even without controls. Additionally, long-term dynamic evaluation of MMP is demonstrated for HeLa cells using **PQ** in oxidative environment. This study is expected to give impetus to the development of mitochondria-related disease diagnosis and drug screening.

## 1. Introduction

The mitochondrial membrane potential (MMP, ΔΨ_mito_) provides the charge gradient required for the maintenance of stable intracellular bioactivities such as ATP production [1,2], regulation of ROS formation [3,4,5], and Ca^2+^ sequestration [6,7,8]. The fluctuation of MMP is a key indicator of mitochondrial dysfunction which is closely related to diseases [9,10,11,12]. Cell inflammation, apoptosis, mitophagy, and canceration can cause abnormality and variation of MMP [13,14,15,16,17,18,19]. For example, mild inflammation triggers a slight MMP decrease [13]; MMP dissipates gradually in the early stage of apoptosis [14]; cell canceration can induce unusually high MMP [16,17]. Therefore, precise monitoring of the change in MMP precisely is crucial to life science, pathology, as well as drug development.

Most of the fluorescent probes for MMP such as commercial 5,5,6,6′-tetrachloro-1,1′,3,3′ tetraethylbenzimi-dazoylcarbocyanine iodide (JC−1) and rhodamine 123 (R123) are attracted to the electronegative interior of mitochondria in live cells and rely solely on the fluorescent intensity of single or dual channels to indicate the MMP level [20,21,22,23,24,25]. However, the fluorescent intensity can be easily affected by the poor photostability and imaging parameters leading to inaccurate determination of the potentials [23,24,25]. Hence, strict additional controls are needed in the process and long-term dynamic monitoring of MMP is difficult using these probes. Moreover, the wide range of fluorescence intensities under a series of MMP changes are hardly detectable in the same parameter. Therefore, fluorescent probes with responsive fluorescence for long-term precise and dynamic monitoring of MMP are rare, albeit highly desired. Ratiometric migratory MMP probes can potentially overcome the influence of imaging parameters on the premise of excellent specificity and resistance to photobleaching and microenvironment interference. Currently, probes with resistance to these interferences remain to be explored [26,27].

In this work, based on the RNA selectivity of pyrrolyl vinyl cation [28,29] and the mitochondrial targeting of lipophilic cation [22], we design pyrrolyl quinolinium cation (**PQ**) to explore a balance between mitochondrial targeting and RNA selectivity. The MMP-responsive fluorescent probe **PQ**, which can migrate reversibly between mitochondria and nucleoli driven by MMP, is demonstrated for dynamic MMP evaluation. As illustrated in Figure 1A, the relative fluorescence intensities of cytoplasm and nucleolus are controlled by MMP with reversible migration of **PQ**, and the ratio can be used to indicate MMP. Under high MMP in live, intact cells, the probe **PQ** concentrates in mitochondria (100~1000× or more compared to the extracellular medium (1×)) following the Nernst equation:ΔΨ = (*RT*/n*F*) × ln(*C*_in_/*C*_out_),
where *R* is the universal gas constant, *T* is the temperature (K), n is the valence of the charged probes, *F* is Faraday’s constant, *C*_in_ and *C*_out_ are the concentrations of the probes on both sides of the charged membrane with a potential ΔΨ [22,30] and emits strong fluorescence (Figure 1C). On the contrary, the concentration of **PQ** in the cytosol is very low (5~10×) and nucleoli fluorescence is weak. With decreasing MMP, the probe translocates from mitochondria to cytosol gradually and binds with RNA in nucleoli due to the intrinsic RNA selectivity accompanied by enhanced fluorescence from nucleoli (Figure 1D). Because mitochondria are located in the cytoplasm, the mean fluorescence intensity ratio of cytoplasm and nucleolus is positively correlated with MMP. The fluorescence ratiometric method avoids the adverse effects of photostability and imaging parameters, endowing **PQ** with the capability of long-term dynamic evaluation of MMP.

## 2. Materials and Methods

### 2.1. Materials and Instruments

2-aldehyde pyrrole, 4-methyl quinoline, fluorescein, and *yeast* RNA were purchased from Aladdin (Shanghai, China), and calf thymus DNA (*ct*DNA) was obtained from Solarbio (Beijing, China). miRNA-21 (5′-UAG CUU AUC AGA CUG AUG UUG A-3′) was synthesized by Sangon Biotech (Shanghai, China) and Iodomethane was bought from Innochem (Gyeonggi, Korea). Piperidine was purchased from Sinopharm Chemical Reagent Co., Ltd. Shanghai, China. All the chemicals were used without purification. Dulbecco’s modified Eagle’s medium (DMEM), 0.25% trypsin/EDTA, penicillin, streptomycin, fetal bovine serum (FBS), and phosphate-buffered saline (PBS) were purchased from Gibco (Calsbad, CA, USA). The water used in the study was purified by a Milli-Q water purification system. All the other reagents were analytical grade. MitoTracker Deep Red FM (MTDR) was purchased from Shanghai Maokang Co., Ltd., Shanghai, China. HeLa cell line was provided by the Type Culture Collection of the Chinese Academy of Sciences, Shanghai, China.

The ^1^H and ^13^C NMR spectra were acquired on a Bruker AVANCE III 400M spectrometer (Billerica, MA, USA) and the high-resolution mass spectra (HRMS) were obtained on a Thermo Scientific^TM^ Q Exactive^TM^ Focus mass spectrometer. The UV–vis absorption spectra were obtained using a quartz cuvette with a 1 cm path length on a TU-1810 spectrophotometer, Persee. The photoluminescence spectra were recorded on the fluorescence spectrophotometer (F-380A, GANGDONG) and confocal fluorescence imaging was conducted on Leica TCS SP8 confocal laser scanning microscope (CLSM, Wetzlar, Germany). The cytotoxicity was determined on a microplate reader (Multiskan GO, Thermo Scientific, Waltham, MA, USA).

### 2.2. The Synthesis of **PQ**

(*E*)-4-(2-(1*H*-pyrrole-2-yl)vinyl)-1-methylquinolinium iodide (**PQ**) was prepared according to Appendix A and 1,4-dimethyl quinoline (II) was synthesized according to the protocol described previously [31]. Amounts of 0.285 g of II and 0.176 g of 2-aldehyde pyrrole (I) were stirred and dissolved in methanol. After a few drops of piperidine were added, the color of the solution turn from light yellow to red rapidly. The mixture was refluxed for 24 h and after cooling slowly, the dark red precipitate was filtered and dried in vacuum with a yield of 65%. The product is analyzed by ^1^H NMR, ^13^C NMR, and HRMS as shown in Appendix A. ^1^H NMR (400 MHz, DMSO-*d*_6_) δ (ppm): 12.00 (s, 1H), 9.11 (d, *J* = 8.0 Hz, 1H), 8.79 (d, *J* = 8.0 Hz, 1H), 8.36 (d, *J* = 8.0 Hz, 1H), 8.30 (d, *J* = 8.0 Hz, 1H), 8.23 (t, *J* = 8.0 Hz, 1H), 8.12 (d, *J* = 6.0 Hz, 1H), 8.04 (t, *J* = 8.0 Hz, 1H), 7.94 (d, *J* = 16.0 Hz, 1H), 7.36 (s, 1H), 6.80 (d, *J* = 4.0 Hz, 1H), 6.34 (s, 1H), 4.40 (s, 3H). ^13^C NMR (400 MHz, DMSO-*d*_6_) δ (ppm): 153.31, 147.38, 139.26, 135.11, 133.87, 131.08, 129.14, 126.82, 126.11, 125.96, 119.76, 118.65, 113.91, 112.36, 111.84, 44.51. HRMS calculated, 235.12; found, 235.12.

### 2.3. RNase and DNase Digestion Experiments

The HeLa cells cultured on confocal dishes were pretreated according to the following procedures. The cells were fixed with 4% paraformaldehyde for 30 min and permeabilized by 0.5% Triton X-100 for 2 min at room temperature. After rinsing twice with PBS, PBS (control), 100 U/mL DNase, and 25 mg/mL DNase-Free RNase were added separately, and the three sets of cells were incubated at 37 °C for 2 h. Then, the three sets of prefixed HeLa cells were stained with 10 μM **PQ** in PBS for 30 min prior to CLSM.

### 2.4. Colocalization of **PQ** and MTDR

The HeLa cells cultured on confocal dishes in advance were incubated with 0.5 μM MTDR for 30 min at 37 °C. After MTDR incubation, the cells were washed twice with PBS and incubated with 10 μM **PQ** in DMEM for another 30 min. Afterward, the cells were rinsed and kept in fresh DMEM or incubated in 2–100 μM CCCP for 10 min. The cell images were acquired on CLSM (λ_ex_/λ_em_ = 638 nm/660–750 nm for MTDR, λ_ex_/λ_em_ = 488 nm/550–650 nm for **PQ**).

### 2.5. MMP Decrease Assay Using CCCP in Real-Time

The HeLa cells were incubated with 10 μM **PQ** in DMEM for 30 min at 37 °C. Afterward, the cells were washed with PBS and kept in fresh DMEM in darkness. Then, the confocal dish was clamped on CLSM and cell images were acquired using a 63× oil immersion objective before and after adding 2, 10, 20, and 100 μM CCCP in darkness successively (λ_ex_/λ_em_ = 488 nm/550–650 nm). Finally, the 100 μM CCCP was removed and replaced by fresh DMEM. The imaging was continued for next 6 min.

### 2.6. Quantitative Analysis for **PQ** Translocation from Mitochondria

The control blank cells and **PQ**-stained live cells (10 μM, 30 min) before and after fixation were prepared as 1.0 × 10^7^ per group. Then, the mitochondria were isolated individually according to the instruction of mitochondria isolation kit and placed in lysates. The absorption spectra of the mitochondrial lysates were obtained on UV–vis absorption spectrophotometer, and the absorbance value at 486 nm was proportional to the concentration of **PQ**.

### 2.7. The Cytotoxicity of **PQ**

HeLa cell aliquots (100 μL, 1 × 10^4^ cells) were seeded on a 96-well plate. After incubation for 12 h, the cells were incubated in various concentrations (0–100 μM) of **PQ** in DMEM for 0.5 h. Cells added with 1% DMSO served as control. Subsequently, the **PQ** solution was removed, and the cells were incubated in fresh culture medium at 37 °C for another 24 h. Then, the treated cells were washed with PBS, added with 0.5 mg/mL MTT in DMEM, and incubated for another 4 h. Then, after the formed formazan crystals were dissolved, the absorbance was measured at 570 nm.

### 2.8. Image Analysis

The images in Appendix A were used to explain the calculation method for the mean fluorescent intensity ratio of cytoplasm and nucleolus (*I*_cytoplasm_/*I*_nucleolus_). In fluorescent and bright field images, the whole cell inside the cell membrane was sketched by the big blue circle (ROI 1) and the total fluorescent intensity (*F*_1_) and the area (*S*_1_) were obtained in ROI 1. The nucleus was sketched by the small blue circle (ROI 2) and the total fluorescent intensity (*F*_2_) and the area (*S*_2_) were obtained in ROI 2. The nucleolus was sketched by the yellow circle (ROI 3) and the mean fluorescent intensity (*I*_nucleolus_) was obtained in ROI 3. The mean fluorescent intensity of cytoplasm (*I*_cytoplasm_) was calculated according to the equation: *I*_cytoplasm_ = (*F*_1_ − *F*_2_)/(*S*_1_ − *S*_2_). The value of *I*_cytoplasm_/*I*_nucleolus_ was the mean fluorescent intensity ratio of cytoplasm and nucleolus.

## 3. Results

### 3.1. Photophysical Properties

**PQ** exhibited a broad absorption band between 400 and 550 nm and the emission band located between 550 and 650 nm as shown in Appendix A. It showed intramolecular charge transfer characteristics [32,33]. With the increase in the solvent polarity (from *n*-BuOH to water), the absorption peak blue-shifted from 505 nm to 465 nm and the fluorescence weakened. Meanwhile, the quantum yield diminished from 2.2% to 0.13% (Appendix A). The fluorescence of **PQ** in the water/glycerol enhanced significantly with the increase in the mixture viscosity (Appendix A). The quantum yield of **PQ** reached 12% in 99% glycerol. On the other hand, **PQ** showed increasing fluorescence when concentrated in water and no effect of aggregation caused quenching (ACQ) (Figure 1C).

### 3.2. Colocalization with MTDR under Various MMP

To verify mitochondria localization and response to MMP, co-staining with commercial mitochondria probe MTDR was carried out with different concentrations of carbonyl cyanide-m-chlorophenylhydrazone (CCCP, a protonophore that can collapse MMP) [31,34,35] to tune the MMP of live cells (Figure 2). In live cells without CCCP, the fluorescence of **PQ** overlapped with that of MTDR, and the Pearson correlation coefficient (*R*_r_) is 0.90. As the CCCP concentration increased from 10 to 20 and 100 μM, nucleoli fluorescence became stronger, but cytoplasm fluorescence was weaker. *R*_r_ decreased to 0.69, 0.30, and 0.12, respectively, indicating that **PQ** molecules tend to escape from mitochondria when MMP decreases. In order to quantify the translocation of **PQ**, the mitochondria were isolated from **PQ**-stained cells before and after fixation and the absorption spectra of mitochondrial lysates were measured (Appendix A). Compared with live cells with high CCCP, the absorbance of **PQ** in mitochondria isolated from fixed cells with vanished CCCP decreased by ~81%, which demonstrates that most **PQ** molecules left mitochondria once MMP was dissipated. The ratio of mean fluorescence intensity of cytoplasm and nucleolus (*I*_cytoplasm_/*I*_nucleolus_) was calculated, and it decreased distinctly with the drop of MMP caused by the increase of CCCP concentration (Figure 2B).

### 3.3. RNA Selectivity

Translocation of **PQ** to nucleoli is believed to arise from the interactions with RNA and therefore, the interactions of **PQ** with nucleic acids, protein, and other biomolecules were investigated. After the addition of RNA, the absorption peak of **PQ** red-shifted from 465 to 504 nm (Figure 3A) and the quantum yield of **PQ** increased from 0.13% to 5.6% by 43 times (Appendix A). Correspondingly, fluorescence increased by 44 times. In contrast, **PQ** exhibited weak fluorescence with other biomolecules including DNA, proteins, amino acids (AAs), and H_2_O_2_ (Figure 3B and Appendix A). The binding mode between **PQ** and RNA was determined by circular dichroism (CD) and the induced CD signal can indicate groove binding between dyes with nucleic acids [36]. As shown in Appendix A, a negative cotton effect was observed, indicating that **PQ** bound with RNA in minor groove binding mode. Furthermore, fluorescence from the **PQ**-stained cells disappeared after RNA digestion, but remained unchanged when DNA was digested (Figure 3C). The results confirm that **PQ** translocates specifically to RNA in cells with smaller MMP.

### 3.4. Dynamic and Precise MMP Evaluation of Cells with CCCP Treatment

To confirm the capability of **PQ** to evaluate MMP precisely, the cells were monitored in real-time. A series of concentrations of CCCP (2, 10, 20, and 100 µM) were introduced to live HeLa cells to induce various MMP situations and fluorescence was monitored for 30 min throughout the entire process (Figure 4A). As shown in Figure 4B, after the addition of 2 µM CCCP, MMP decreased slightly and the mean fluorescence intensity ratio of cytoplasm and nucleolus *I*_cytoplasm_/*I*_nucleolus_ decreased from 4.0 to 3.1 in 6 min. When the concentration of CCCP was increased to 10 µM and 20 µM, the *I*_cytoplasm_/*I*_nuleolus_ ratios continued to decrease to 1.4 and 0.7 sharply. When CCCP was in excess (100 µM), MMP tended to vanish, and the ratio decreased to the smallest value ~0.50. Theoretically, MMP recovers to a certain extent after removing all the CCCP. Experimentally, the cytoplasm fluorescence recovered but nucleoli fluorescence diminished again. The *I*_cytoplasm_/*I*_nucleolus_ ratio increased rapidly to 3.0 (Figure 4B), demonstrating that the mean fluorescence intensity ratio of cytoplasm and nucleolus is positively correlated with MMP and various MMP situations can be semiquantitatively evaluated.

### 3.5. Photostability and Cytotoxicity

Fluorescent probes for evaluating MMP such as rhodamine 123 (R123) rely on fluorescence intensity [21,22] and it is somewhat limited because strict positive and negative controls are needed to evaluate the potential range of cell samples. The poor stability influences the evaluation accuracy. Thus, the stability of **PQ** was investigated under light irradiation, various pH, and in presence of oxidizing agent (H_2_O_2_) and reducing agent (vitamin C, VC). After continuous light irradiation for 60 min, **PQ** maintained 83% fluorescence in aqueous solution and 74% fluorescence in RNA solution (Appendix A). There was no significant change in fluorescence of **PQ** in the pH range of 3~11 (Appendix A), and in 0.5–2 mM H_2_O_2_ and 0.5–5 mM VC solution, the fluorescence showed no obvious fluctuation (Appendix A). Moreover, **PQ**- and R123-stained cells were imaged 50 times successively (Figure 5). **PQ** retained 89% of the original fluorescence intensity and R123 remained 75%, indicating that **PQ** has better photostability but R123 is not suitable for long-term MMP monitoring. Although the fluorescence intensity in the 50 images decreased slightly, the mean fluorescence intensity ratios of cytoplasm and nucleolus remained constant nearly (Figure 5B). CLSM images of the same cells excited by a series of laser intensities were also obtained and the corresponding *I*_cytoplasm/_*I*_nucleolus_ remained almost unchanged (Appendix A). All the results demonstrate that the MMP evaluation method based on the fluorescence intensity ratio eliminates the negative effects caused by photostability and excitation conditions consequently boosting the accuracy. Furthermore, the MTT assay indicated negligible cytotoxicity even at a high concentration of 50 µM (Figure 5C) further corroborating the suitability for long-term monitoring.

### 3.6. Dynamic MMP Monitoring in Oxidative Environments

The rising of reactive oxygen species (ROS) level is an important factor causing cell inflammation and apoptosis as well as an inducer of many diseases [37,38]. To confirm the ability of **PQ** to monitor MMP in real time further, H_2_O_2_ (0.5 mM and 2 mM) was used to create an oxidative environment in which **PQ**-stained live cells were subjected to fluorescence imaging. As shown in Figure 6, in a low concentration of H_2_O_2_ medium (0.5 mM), the ratio decreased gradually from 3.5 to 1.3 for cell “c”; whereas, in the high concentration medium (2 mM) the ratio decreased sharply in 1 min from 3.9 to 0.9 for cell “d”, and then continued to decrease to 0.6 gradually. The results disclose that the MMP of live cells descends on a small scale at low ROS levels but drops sharply at high ROS levels until it vanishes.

MMP is different from cell to cell and its fluctuation reflects mitochondrial activities including electron transport, oxidative phosphorylation, glucose intake, and ROS production [12,22,39]. Therefore, the detection and evaluation of MMP are essential to life science and other biological fields [22]. Unfortunately, previous works have mostly focused on the determination of mean MMP values at selected time points, but in situ long-term monitoring has rarely been performed quantitatively. With the development of the probe **PQ** and the fluorescence intensity ratio strategy described in this paper, real-time monitoring of MMP in situ can be accomplished precisely. This probe facilitates the biological mechanism study of apoptosis and autophagy in addition to contributing to pathology research and the development of novel drugs.

## 4. Conclusions

In conclusion, we have synthesized a reversibly migratable fluorescent probe **PQ**. The distribution of **PQ** in mitochondria and nucleoli is controlled by MMP. Under high MMP, most molecules accumulate in mitochondria and emit strong fluorescence. With decreasing MMP, the molecules are distributed in both mitochondria and nucleoli, and both emit moderate fluorescence. When MMP dissipates, most molecules bind with RNA and the nucleoli emit fluorescence, but cytoplasm fluorescence is weak. By monitoring the fluorescence intensity ratio of cytoplasm and nucleolus, MMP can be evaluated precisely in various MMP situations with good photostability and negligible cytotoxicity. Additionally, the MMP of live cells in protonophores and oxidative environments has also been monitored long-term. This probe is expected to have broad application prospects in mitochondria-related life science, pharmaceutics, and medical science.

## Figures and Tables

**Figure 1 biosensors-12-00798-f001:**
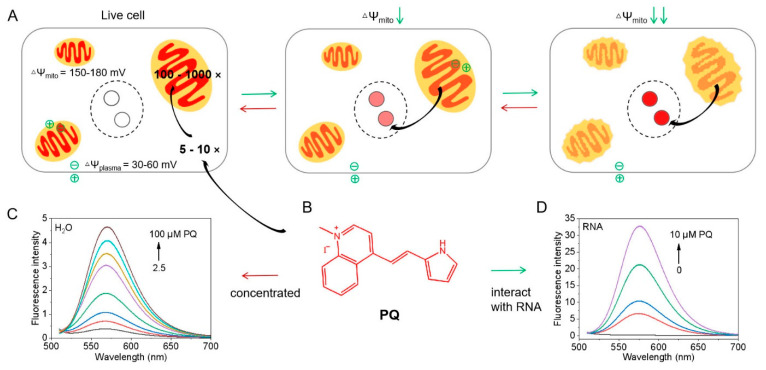
Principle of the reversibly migratable fluorescent probe **PQ** for the evaluation of MMP. (**A**,**B**) Schematic illustration of **PQ** migration between mitochondria and nucleolus and the chemical structure of **PQ**; (**C**) fluorescence spectra of **PQ** for increasing concentrations (2.5, 5, 10, 20, 40, 60, 80, 100 µM from bottom to top) in aqueous solution mimicking high concentrations in mitochondria with high MMP; (**D**) the fluorescence spectra of **PQ** at relatively low concentrations (0, 1, 2.5, 5, 10 µM from bottom to top) in 0.5 mM *yeast* RNA, mimicking the interaction between **PQ** and RNA in nucleolus under decreasing MMP.

**Figure 2 biosensors-12-00798-f002:**
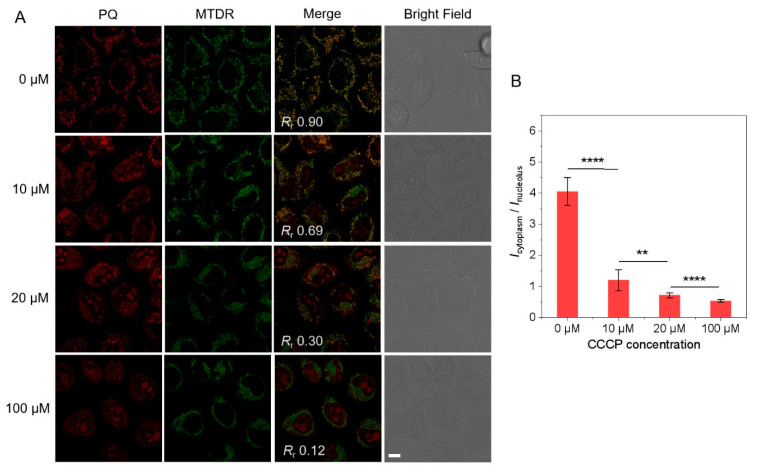
CLSM images of **PQ** in HeLa cells with decreasing MMP. (**A**) Colocalization images of live HeLa cells co-stained with **PQ** and MTDR after treatment with different concentrations of CCCP (0, 10, 20, 100 µM) for 10 min (λ_ex_/λ_em_ = 488 nm/550–650 nm for **PQ**, λ_ex_/λ_em_ = 638 nm/660–750 nm for MTDR, scale bar: 10 μm) and (**B**) Corresponding ratios of relative fluorescence intensities in the cytoplasm and nucleolus (*I*_cytoplasm_/*I*_nucleolus_). [Mean ± SD (n = 50); **, *p* < 0.01; ****, *p* < 0.0001; Student’s *t*-test].

**Figure 3 biosensors-12-00798-f003:**
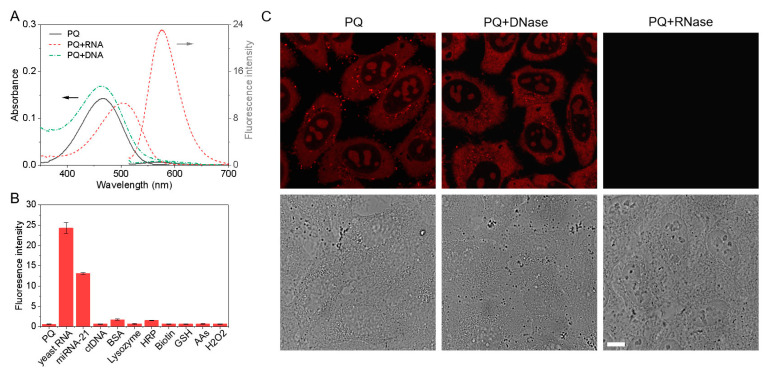
RNA selectivity. (**A**) UV–vis absorption and fluorescence spectra of 5 µM **PQ** in the presence of RNA/DNA. The black arrow means “using the left coordinate axis”, and the grey arrow means “using the right coordinate axis”; (**B**) Fluorescence comparison of 5 µM **PQ** in the presence of *yeast* RNA (1.0 mM phosphate), miRNA-21 (48 µM phosphate), *ct*DNA (1.0 mM phosphate), proteins (1.0 mg/mL), biotin (10 mM), glutathione (GSH, 10 mM), amino acids (10 mM) and H_2_O_2_ (10 mM) in PBS. The proteins include bovine serum albumin (BSA), lysozyme (Lyso), and horseradish peroxidase (HRP). The amino acids (AAs) include homocysteine (Hcy), cysteine (Cys), glycine (Gly), alanine (Ala), norvaline (Nor), glutamic acid (Glu), histidine (His), isoleucine (Iso), leucine (Leu), methionine (Met), phenylalanine (Phe), proline (Pro), serine (Ser), threonine (Thr), tryptophane (Try), and tyrosine (Tyr). Data represent mean ± SD (n = 3); (**C**) CLSM images of **PQ** in the fixed HeLa cells with or without DNase/RNase digestion (λ_ex_/λ_em_ = 488 nm/550–650 nm, scale bar: 10 μm).

**Figure 4 biosensors-12-00798-f004:**
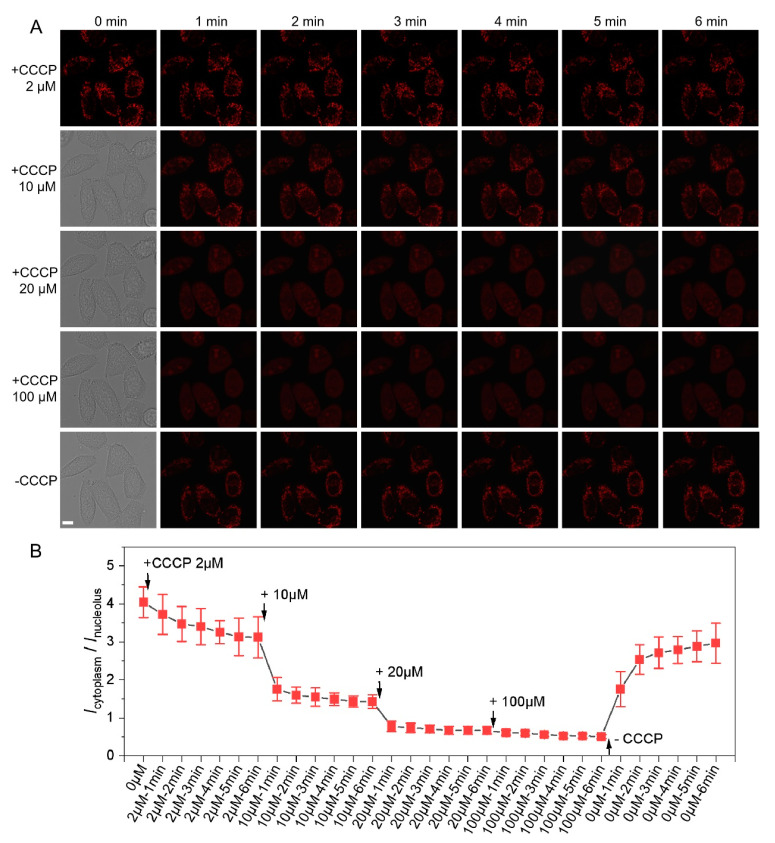
Dynamic MMP monitoring. (**A**) CLSM imaging of live HeLa cells stained with 10 µM **PQ** when incubated continuously in 2–100 µM CCCP for 0–6 min (λ_ex_/λ_em_ = 488 nm/550–650 nm, scale bar: 10 μm) and (**B**) Corresponding *I*_cytoplasm_/*I*_nucleolus_ changes, data represent mean ± SD (n = 5).

**Figure 5 biosensors-12-00798-f005:**
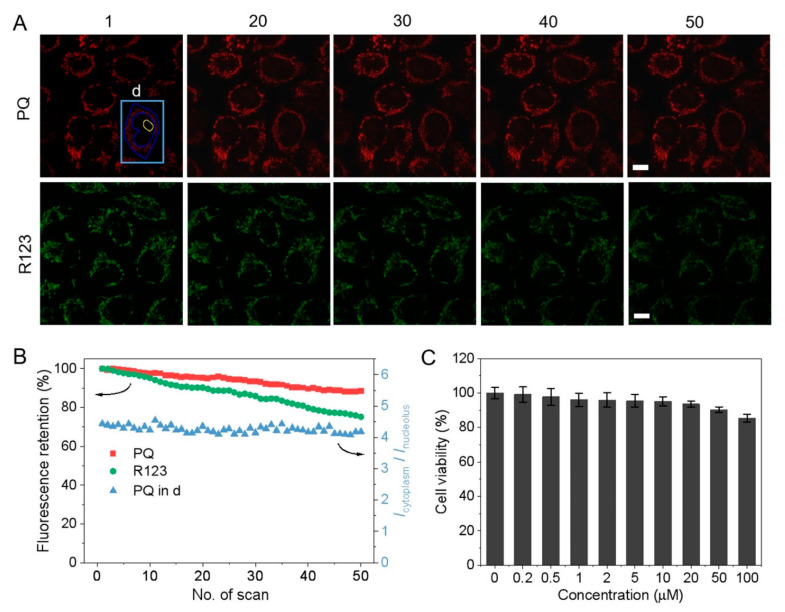
Photostability and cytotoxicity of **PQ**. (**A**) CLSM images of **PQ** and R123 in live HeLa cells with continuous scanning (λ_ex_/λ_em_ = 488 nm/550–650 nm for **PQ**, λ_ex_/λ_em_ = 488 nm/500–550 nm for R123, scale bar: 10 μm); (**B**) mean fluorescence intensity changes of **PQ** and R123 in live HeLa cells and I_cytoplasm_*/*I_nucleolus_ changes of the cell “d” marked in (**A**) with continuous scanning; (**C**) viability of HeLa cells incubated with different concentrations of **PQ** for 24 h in darkness, the data represent mean ± SD (n = 5).

**Figure 6 biosensors-12-00798-f006:**
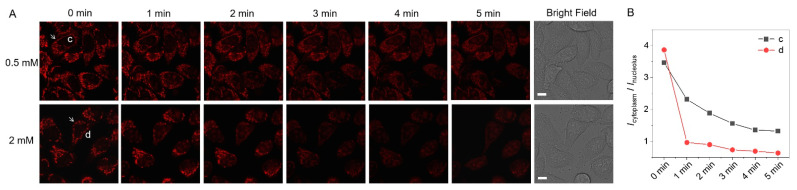
Dynamic MMP monitoring in oxidative environments. (**A**) CLSM imaging of live HeLa cells stained with 10 µM **PQ** before and after being placed in 0.5 and 2 mM H_2_O_2_ for 0–5 min (λ_ex_/λ_em_ = 488 nm/550–650 nm, scale bar: 10 μm) and (**B**) corresponding *I*_cytoplasm_/*I*_nuleolus_ changes of the cells “c” and “d” marked by white arrow in (**A**).

## Data Availability

Original and raw data files are available from the authors upon reasonable request.

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
