# Peer review of "Reversibly Migratable Fluorescent Probe for Precise and Dynamic Evaluation of Cell Mitochondrial Membrane Potentials"

_biosensors, 2022, doi:10.3390/bios12100798_

Round 1

Reviewer 1 Report

Manuscript: biosensors-1914019

The manuscript is entitled “Reversibly migratable fluorescent probe for precise and dynamic evaluation of cell mitochondrial membrane potentials”.

In this work, the authors designed and synthesized MMP-responsive fluorescent probes that reversibly migrate between mitochondria and nucleoli. The probe avoids the negative effects of photostability and imaging parameters and monitors MMP conditions in real time. However, we think there are still some issues in this manuscript need to be corrected by the author before publication.

(1)      The probes should be evaluated for in vitro stability, such as time, temperature, pH.

(2)      When performing biological imaging experiments, the emission wavelength range of the imaging photos should be marked in a ma.

(3)      The words and grammar of the article will be further modified and polished.

(4)      Some representative literatures should be cited in the background, Such as: Chinese Chemical Letters 32 (2021) 3895–3898,Chemical Engineering Journal 437 (2022) 135397,Sensors and Actuators: B. Chemical 370 (2022) 132433.

Reviewer 2 Report

In this submitted article, Guofen Song et al. reported a mitochondrial membrane potential (MMP)-responsive fluorescent probe, PQ, which can determine the dynamic evaluation of MMP through reversible migration between mitochondria and nucleolus. The authors demonstrated PQ translocation from mitochondria to nucleoli when CCCP was treated. The dynamic change of MMP was evaluated by the fluorescence intensity ratio of cytoplasm to the nucleolus. The authors claimed that long-term dynamic evaluation of MMP is also possible using PQ in an oxidative environment. The key concept of this work is good, and a few evidence are supported the author’s key claim. However, a few discrepancies in the manuscripts may lead to confusion. I recommend acceptance after major corrections, as mentioned below points.

General concerns and questions:

1)      In Figure 2, after variable CCCP concentrations treatment, the PQ translocates from mitochondria to the nucleolus. Surprisingly, there was no effect on Mitotracker deep red (MTDR), a mitochondrial potential-dependent dye (DOI: 10.3389/fncel.2016.00076). The staining of the MTDR is not expected after CCCP treatment as MMP has changed. This observation leads to confusion. Please clarify your observations and conclusion from this experiment.

2)      In Figure 3, the authors explained that PQ binds with RNA at the nucleolus. Thus, after treatment of RNase, the signal intensity should be disappeared in the nucleolus, and the mitochondrial signal should be minimally affected. However, it is confusing how the signals disappeared completely from mitochondria.

3)      In Figure 4, the experiment should be clearly described to understand better for the readers. Especially how the washing was performed and how to get the images of precisely the same cells after washing. It is also not clear how the MMP has been restored within 3 min. 

4)      In Figure 5, it is recommended to show the area of cytoplasm and nucleolus for measurement of (Icytoplasm / Inuleolus) changes. It isn't easy to follow the way it is presented in Figure 5.  

5)      The most relevant articles should also be cited such as; a) doi.org/10.2144/000113610; b) doi: 10.21769/BioProtoc.3128; c) doi.org/10.1016/S0006-3495(99)77214-0; d) doi: 10.1021/acs.analchem.2c02177; e) doi: 10.3389/fncel.2016.00076

6)      Minor points: There are few typo errors need to look carefully.

a.       Page 4, Line 157, “drop of MMP caused by the drop of MMP caused by the increase of CCCP (Figure 2B)” should be corrected.

b.      Page 5, Line 173, “CD signal can indicate groove binding between dyes with nucleic acids [27]. As shown in Fig. 3C”. There is no CD spectra in Figure 3C.

Reviewer 3 Report

A pyrrolyl quinolinium (PQ) probe was developed for mitochondria and RNA sensing. By changing the mitochondrial membrane potential (MMP), the fluorescent signal location changed between mitochondrial and nucleoli. Overall, this work is interesting, however, two key questions remain unanswered.

1. It is expected that PQ fluorescence may be affected by many factors. The authors showed the influence of solvent polarity and viscosity [Sensors and Actuators B: Chemical 2022, 132003; Molecules 2016, 709]. Due to the component of pyrrolyl, PQ is believed to be pH sensitive, please include pH-dependent optical properties. Thus, the low pH environment in mitochondria should not be ignored.

2. The authors conclude that ‘the distribution of PQ in mitochondria and nucleoli is controlled by MMP’, which was evidenced by the fluorescence signal switch. However, there is no direct proof of PQ translocation. In other words, the distribution of PQ may be stable, just the microenvironment affected by MMP ‘turn-on’ and ‘turn-off’ the ones in mitochondria and nucleoli in a different way.  

Reviewer 4 Report

In this manuscript, Song et al synthesized and tested a fluorescent probe, PQ, as ratiometric sensor for mitochondrial membrane potential (MMP) in live cells. PQ enters the mitochondria by its positive charge. As the MMP decreases, it migrates to the nucleolus due to its RNA binding affinity, allowing for ratiometric (yet only qualitative) evaluation of MMP in cells. The probe is validated by treatments that depolarize MMP, including CCCP and H2O2. 

Sensing MMP in live cells has potentials in biomedical and clinical research. Most experiments are appropriately designed and the results are reasonably interpreted. However, the lack of discussion/comparison of existing ratiometric MMP probes leaves the novelty and significance of this work in question (see major point #1 below). Some experiments need biological replicates with better controls.

Major points:

1.     There is no mentioning of existing ratiometric, especially migratory MMP probes in the manuscript. The authors should discuss and compare PQ with existing MMP probes, such as the widely used JC-1, LAD in Li et al (2017), MR1/2/3 in Tian et al (2019), and Cz-Bz-1 in Li et al (2019). Specifically, what new opportunities would PQ offer compared to these existing probes? Will PQ open doors to new questions that previous probes cannot? Is PQ better at optical properties? These questions need to be addressed to allow fair evaluation of the manuscript’s novelty and contribution to the field.

2.     The dynamic MMP monitoring experiment (Figure 4) is confusing and it seems that it was not repeated. The intensity ratio starts at 3, while in Figure 2B the average of normal intensity ratio is 4, suggesting the example in Figure 4 is not representative. During the increase of CCCP concentration, each concentration was only monitored for ~3 minutes, so that the intensity ratio had not reached a steady state before the next concentration was added. Consequently, the intensity ratios with different CCCP concentrations do not match those shown in Figure 2. The authors should redo this experiment with longer incubation periods and show the data of multiple cells.

3.     The organization of sections is a bit random. In vitro and in vivo results, and dye characterization and application are mixed up. It would benefit from rearranging the sections such that the authors start with dye characterization and move on to live-cell applications to improve readability.

Minor points:

1.     Line 24-25: the final sentence is too unspecific. Need something more relevant to the probe and method itself.

2.     Line 60: how did the authors get 5-10x? 

3.     How is fluorescent intensity reported throughout the figures? If the values are the raw readings from the spectrometer, the fluorescence seems to be rather dim. 

4.     Line 83: “Gibbco” should be “Gibco”.

5.     Image analysis is missing from the Methods section, including how the authors define the I, how the correlation coefficients were calculated. 

6.     Experimental details of R123 are missing. If the excitation conditions of R123 are different from PQ in Figure 5, the photostability comparison will not be very meaningful.

7.      Some figures are missing scale bars or the scale bars are undefined.

8.     How are the P values defined? What are the sample sizes and tests?

9.     Line 168-169, switch “fluorescence” and “quantum yield”, as the QY is the cause, and the fluorescence is the consequence.

10.  The CD spectrum shows only wavelengths >350 nm. Typically the peaks of RNA/DNA are between 200-350 nm. It is unclear what the authors expect here. Any positive controls?

11.  Figure 4: why are there bright-field images rather than fluorescent images for some time points?

12.  Figure 5C: the assay lacks a positive control. It could also be that the assay is not working appropriately.

13.  Figure 6A: there seems to be bleaching of the dye at 2mM H2O2? Is it optical or chemical?

14.  Line 247-248: “in situ long-term monitoring has rarely been performed” is not true. There are probably hundreds of papers doing this over the past few decades.

15.  Reference 14-15: the journal names should be italic.

Reviewer 5 Report

The manuscript by Song and others describes a fluorescent sensor, pyrrolyl quinolinium (PQ), which detects mitochondrial membrane potential (MMP). In normal cells, PQ is localized in mitochondria because of its cationic nature, and so the fluorescence is low in cytosol or in nucleus. However, when MMP is decreased, PQ is released from mitochondria and then moves to nucleolus because of its intrinsic affinity toward RNA. Hence, by measuring the fluorescence ratio of PQ between nucleolus and cytosol, researchers can semi-quantitatively measure MMP in a real time manner.

Although there are fluorescent sensors for MMP, such as rhodamine 123, most of them are intensiometric sensors that only change fluorescence intensity in one site at one wavelength. On the other hand, detection using PQ has a ratiometric nature and it is resistant to photobleaching or dye leakage. The authors demonstrated the utility of this sensor by imaging dynamic change of MMP induced by oxidative stress or CCCP treatment.

Considering the good performance and unique mechanism of PQ in cell imaging, I believe the manuscript should be accepted in Bisensors after minor revisions. The detailed comments are as follows.

1.    The authors should explain about the design of the compound, PQ. Why did they focus on this particular structure?

2.    Line 53, 59: When the authors use the expression such as “100~1000×”, please explain what is 1×. Otherwise, the readers don’t understand the meaning.

3.    The authors should explain how they calculated fluorescence ratio between cytosol and nucleolus. How did they set the regions of interest without relying on any organelle markers? Did they circle some regions manually?

4.    Line 129: 1×104 cells. “4” should be superscript.

5.    Basically, “Methods” and “Results” section should be written using the past tense.

6.    Figure 2A. Scale bar should be defined.

7.    Figure 2B. The number of trials as well as definition of error bars should be added in the legend.

8.    Figure 4A. Scale bar should be defined.

9.    Figure 5A. Scale bar should be defined.

10.  Figure 6A. Scale bar should be defined.

Round 2

Reviewer 2 Report

The authors have explained all of my queries nicely. The manuscript has improved after revision. Thus, I recommend acceptance of this manuscript.

Author Response

We appreciate the positive comments from the reviewer #2.

Reviewer 3 Report

The revised manuscript is much better and recommended for publication. 

Author Response

We appreciate the positive comments from the reviewer #3. Also, we have spell-checked the manuscript thoroughly and polished the English language further.

Reviewer 4 Report

The authors have sufficiently addressed my comments. I recommend its publication after correction of grammatical errors.

One small point: Line 161: "blue" should be "yellow".

Author Response

We have revised the manuscript according to the comment. The “big blue” have been corrected to “yellow” in section 2.8 of the manuscript. Sorry for this error